# Metacyclogenesis as the Starting Point of Chagas Disease

**DOI:** 10.3390/ijms25010117

**Published:** 2023-12-21

**Authors:** Alessandro Zanard Lopes Ferreira, Carla Nunes de Araújo, Isabela Cunha Costa Cardoso, Karen Stephanie de Souza Mangabeira, Amanda Pereira Rocha, Sébastien Charneau, Jaime Martins Santana, Flávia Nader Motta, Izabela Marques Dourado Bastos

**Affiliations:** 1Pathogen-Host Interface Laboratory, Department of Cell Biology, University of Brasilia, Brasilia 70910-900, Brazil; 2Faculty of Ceilândia, University of Brasilia, Brasilia 70910-900, Brazil; 3Laboratory of Protein Chemistry and Biochemistry, Department of Cell Biology, University of Brasilia, Brasilia 70910-900, Brazil

**Keywords:** *Trypanosoma cruzi*, Chagas disease, metacyclic trypomastigote, protozoan parasite, metacyclogenesis

## Abstract

Chagas disease is a neglected infectious disease caused by the protozoan *Trypanosoma cruzi*, primarily transmitted by triatomine vectors, and it threatens approximately seventy-five million people worldwide. This parasite undergoes a complex life cycle, transitioning between hosts and shifting from extracellular to intracellular stages. To ensure its survival in these diverse environments, *T. cruzi* undergoes extreme morphological and molecular changes. The metacyclic trypomastigote (MT) form, which arises from the metacyclogenesis (MTG) process in the triatomine hindgut, serves as a crucial link between the insect and human hosts and can be considered the starting point of Chagas disease. This review provides an overview of the current knowledge regarding the parasite’s life cycle, molecular pathways, and mechanisms involved in metabolic and morphological adaptations during MTG, enabling the MT to evade the immune system and successfully infect human cells.

## 1. Introduction

Considered a major protozoan Neglected Tropical Disease in Latin America, Chagas disease is caused by the parasite *Trypanosoma cruzi*, predominantly transmitted to humans by triatomine vectors, from the Reduviidae family. Despite that, there are other modes of transmission, including vertical, organ transplantation, blood transfusions from infected individuals, and oral transmission through the ingestion of food contaminated with triatomine faeces [1]. Chagas disease control programmes, initiated in the mid-20th century, have continuously evolved and expanded over decades, notably preventing blood transfusion transmission and reducing new infections in South America via the vectorial route. The duration and strength of these efforts can significantly vary by country and region, particularly in areas once deemed controlled. It is noteworthy that triatomine vectors can re-establish themselves in areas where they were previously eliminated, often due to changes in environmental conditions or public health efforts, and consequently cause a resurgence of infection [2]. Moreover, although the ingestion of contaminated food or beverages is responsible for outbreaks in rural and periurban areas, this transmission causes more severe clinical conditions and higher mortality in relation to percutaneous vector-borne transmission [3]. Despite being considered endemic in Latin America, Chagas disease has become a global disease due to migratory flows, spreading to non-endemic countries in North America, Europe, and Asia, with an estimated 6 to 7 million infected people worldwide [1].

Approximately 40% of infected individuals develop cardiopathy, gastrointestinal disorders (megaoesophagus or megacolon), or both clinical manifestations [1,4]. Cardiac involvement is the main cause of death through dilated cardiomyopathy, congestive heart failure, arrhythmias, cardioembolism, stroke, and acute myocarditis [4]. The current treatment is based on chemotherapy using the nitroheterocyclic compounds benznidazole or nifurtimox that commonly fails to treat the chronic stage of the disease [5,6]. Also, no vaccine is available for Chagas disease, although studies on prophylactic and therapeutic vaccines are currently under development [7]. Due to the difficulty of early infection diagnosis and the lack of treatment leading to chronic disease cure, each affected individual incurs USD 474 in health-care costs and 0.51 disability-adjusted life-years (DALYs) annually, reflecting an estimated global annual burden of USD 627.46 million in health-care costs and 806,170 DALYs [8]. This scenario imposes the urgency of developing effective treatment and prevention alternatives, which require scientific efforts in order to unravel the biological aspects of this parasite during its life cycle.

*T. cruzi* has a complex dixenous life cycle, including epimastigote, metacyclic trypomastigote (MT), amastigote, and bloodstream trypomastigote (BT) as major developmental forms that transit between mammalian hosts and hematophagous invertebrate vectors [9,10]. In the triatomine gastrointestinal tract, the parasite passes through a series of metabolic and morphological changes that result in the MT, the primarily mammalian infective form. This form represents the bridge between its hosts since this transitioning stage found in the invertebrate host faeces is responsible for evading the immune system, entering the vertebrate host cells, and establishing the infection [9,10]. In this review, we explore the metacyclogenesis (MTG) process and the morphological and metabolic features of *T. cruzi* MT and provide insights into molecular pathways, highlighting potential targets for the development of new strategies to interrupt parasite transmission.

## 2. *Trypanosoma cruzi* Presents a Complex Dual-Host Life Cycle

The life cycle of *T. cruzi* has been elucidated for almost a century [11]; however, there is still conflict in the literature regarding some specific details. Discoveries dating back to the early studies, along with recent findings, offer new perspectives on understanding the parasite’s biology and the pathogenesis of the disease. Since Chagas’ studies [11], epimastigotes were considered non-infective forms due to the fact that experimental evidence had shown that, when maintained in axenic cultures, they were unable to infect mice and were highly sensitive to complement, undergoing lysis when incubated in the presence of fresh guinea pig or human serum [12]. However, the observation of epimastigote-like forms inside mammalian cells in culture raised the question of their potential infectivity [13].

The classic version of the *T. cruzi* life cycle involves the mammalian and the triatomine hosts, with four well-elucidated stages that are divided into replicative or infective forms. Vertebrate host infection begins with the non-replicative MT present in the insect faeces, which enter the wound caused by the triatomine bite or through mucous membranes. Following that, MTs enter cells by binding to receptors, forming the parasitophorous vacuole (PV) in a wide range of phagocytic and non-phagocytic nucleated cells, where MTs differentiate into round replicative forms called amastigotes. These forms escape from the PV into the cytoplasm, multiply by binary fission, and transform into trypomastigotes that disrupt the host cell and disseminate into the bloodstream and tissues, where they can infect more cells. Once bloodstream trypomastigotes (BTs) are ingested by the triatomine, they transform into epimastigotes in the vector midgut, where they replicate, migrate to the insect posterior gut, and attach by their flagella to the gut waxy cuticle to differentiate into MTs (Figure 1) [4,14].

However, it has been noted that the traditional life cycle of *T. cruzi* is insufficient to explain the complexity of the process, both in triatomine insects and in mammals. From his initial observations on the aetiology of the disease, Carlos Chagas documented a range of forms giving rise to MTs [11]. Furthermore, he also identified a pleomorphic population of parasites in the blood of infected mammals, primarily consisting of slender and broad BT forms [11]. These forms exhibit variations in infectivity, susceptibility to antibodies, and tissue tropism [17,18]. The ratio between the slender form, which is more commonly observed during early parasitaemia, and the broad forms, which persist in the bloodstream for longer periods, is strain-dependent and may therefore influence the progression of the disease. Additionally, apart from these pleomorphic BT forms, amastigotes (constituting approximately 10%) resulting from either cell lysis or extracellular differentiation of BTs, as well as intermediate forms derived from this process, can be found in the blood, primarily during the acute phase of infection [19].

Studies in different cell types and tissues have demonstrated the presence of distinct non-canonical morphological forms of *T. cruzi* during the amastigote-to-trypomastigote differentiation process [15,20]. The presence of an intracellular epimastigote-like form was described for the first time in tissue culture cells by Meyer and De Oliveira in 1948. Other studies have shown, through fluorescence microscopy, that the transitional forms between amastigotes and trypomastigotes have morphological characteristics similar to epimastigotes [21]. However, it remains to be seen whether these non-classical parasite forms are transitional or intracellular stages and what kind of physiological role they play.

In the insect vector, it was observed that BTs quickly differentiate into amastigote-like forms that, in the posterior midgut (PM), will give rise to infective epimastigotes, a process known as primary epimastigogenesis. The latter migrate to the hindgut, where MTG occurs, and in response to suitable environmental conditions, MTs may perform secondary epimastigogenesis. The signalling mechanisms used by MTs to convert into epimastigotes are probably stimulated by the lack of a carbon source, although further studies are needed to confirm this hypothesis [16].

Both trypomastigotes and amastigotes are capable of infecting vertebrate host cells. Amastigotes display the capacity to infect and efficiently complete their life cycle within both phagocytic and non-phagocytic host cells, as evidenced by findings from in vitro and in vivo (mouse) infectivity studies. In human monocytes, amastigotes initiate replication without delay, while trypomastigotes present a significant delay between invasion and the beginning of DNA duplication [14,22]. Furthermore, Kessler and colleagues [16] demonstrated, for the first time, that newly differentiated epimastigotes (rdEpis) (prior to four cycles of division) are virulent parasites with the ability to infect host cells and exhibit resistance to complement lysis [15,16]. Nevertheless, the authors emphasise that, while this form shares some characteristics with both epimastigotes/amastigotes and trypomastigotes, the virulence of rdEpis cannot be solely attributed to residual protein expression similarities or a ‘molecular memory’ from morphogenesis. It is likely that virulence is influenced by a specific set of rdEpi proteins, making them potential targets for drugs or vaccine candidates [16]. In conclusion, the complex transition from trypomastigotes to epimastigotes underscores the importance of identifying these epimastigote-specific proteins, which could significantly contribute to the development of new therapeutic and preventive strategies.

## 3. Morphological Features of Metacyclic Trypomastigotes

In MT, morphological changes include alterations and repositioning of the nucleus and kinetoplast, a decrease in the reservosome volume, chromatin remodelling, elongation of the body, and reposition of the flagellum in relation to the nucleus [9,14]. In epimastigotes, a disc-shaped kinetoplast containing densely packed kDNA fibres is positioned anterior to a rounded nucleus [9]. In the intermediate I form, the flagellum and the disc-shaped kinetoplast move towards the posterior region of the cell body, lateral to the nucleus, which gets deformed and elongated. In the intermediate II form, the kinetoplast, which is more posterior, appears to be disc-shaped and contains densely packed kDNA and a slightly elongated nucleus. In the intermediate III stage, the epimastigote-like shape is maintained, with the nucleus being more elongated and the flagellum emerging from the posterior region of the cell, where the kinetoplast is localized. Finally, the intermediate III form completes the metacyclogenesis, differentiating into metacyclic trypomastigotes, whose kinetoplast takes on a globular shape filled with loosely arranged kDNA and presents a thinner and more sinuous cell body and nucleus [9] (Figure 1). Studies have also shown that shedding of vesicles from the parasite cytoplasmic membrane also occurs in advanced stages of differentiation, which increases MTG and the susceptibility of mammalian cells to infection [9,14]. However, it is important to notice that morphological alterations during the MTG may vary among strains from different DTUs (Discrete Typing Units) [23].

## 4. Metacyclogenesis: What Is Known So Far

The known necessary conditions to trigger MTG include high osmolarity and the depletion of simple monosaccharides such as glucose and fructose, resulting in nutritional and cellular stress, along with parasite attachment to bug intestinal hindgut cells [24,25]. The hindgut is an environment characterised by a scarcity of nutrients, the presence of cleaved peptides originated from blood feeding, and lower levels of glucose and heme. A reductive environment with molecules as urate and NAC (N-acetyl-cysteine) and pH acidification seems to set mechanisms related to cellular stress and also favours MTG [26,27,28,29]. This environment leads to protozoan attachment, differentiation, and later detachment of the newly formed MTs [25,30,31].

During MTG, the expression of a series of virulence factors, heat shock proteins, nuclear proteins, and many other molecules is reported. Many of them are involved in complement system evading, modulation and/or manipulation of defence cells’ responses, mucosal cell infection, and, in the case of parasite oral ingestion, resistance to the mammalian digestive system [9,32]. However, the exact cellular mechanisms underlying MTG remain partially unanswered. An overview of this complex process that involves cyclic adenosine 3′,5′-monophosphate (cAMP) signalling, autophagy, apoptosis, and nuclear remodelling will be presented below.

*cAMP* is one of the most studied and well-characterised second messengers in eukaryotes and is involved in a multitude of responses through cAMP-dependent signal pathways [33]. Two key enzymes in its metabolism are adenylate cyclase (AC), which converts ATP to cAMP upon activation by upstream signalling, and phosphodiesterase (PDE), which is responsible for the degradation of cAMP [34]. There is a direct correlation between MTG and the increased levels of cAMP in the parasite [35] (Figure 2A). It is thought that GDF, an α^D^-globin-derived peptide, is one of the possible triggers for cAMP increase [36]. This peptide seems to activate a G-protein-coupled receptor expressed by *T. cruzi* epimastigotes [37]. Upon GDF activation, the transduction of this stimulus leads to increased levels of TcAC (*T. cruzi* adenylyl cyclase) and, concomitantly, TcPDE inhibition [38,39]. In agreement, the knockout of pyruvate dehydrogenase phosphatase (TcPDP) inhibits downstream pyruvate dehydrogenase (TcPDH; involved in ATP production) activation and leads to decreased MTG [40].

*Autophagy and apoptosis-like pathways* have been proposed as mechanisms required for effective MTG [41]. Autophagy is employed to recycle damaged cellular components and to provide a means of nutrition in starvation conditions [42]. Nutritional stress leads to a possible increase in spermidine and, thereafter, spermine. These two molecules can potentially inhibit the activity of histone acetyltransferases (HATs), resulting in enhanced transcription of Atg (Autophagy Related Protein) genes, more specifically TcAtg8 and TcAtg4 [43,44]. Both proteins are essential for autophagosome and later autolysosome formation, structures where autophagy is conducted. Inhibition of the target of the rapamycin (mTOR) protein and activation of the Vps34 pathway were also observed during MTG (Figure 2A) [44]. mTOR and Vps34 have opposite effects regarding autophagy. mTOR is a well-studied pathway related to cellular survival that is sensitive to nutritional fluctuations and able to induce glycolysis and inhibit autophagy. Vps34 induces the production of phosphatidylinositol 3-phosphate (PI3P), a phospholipid required for membrane formation and therefore necessary for membrane modulation and autophagosome formation [44,45].

Cruzipain, a cathepsin L-like cysteine protease detected in all stages of the parasite cell cycle [46], localises within the reservosomes and presents several functions on the homeostasis of the parasite, such as reservosome consumption, thus positively affecting MTG (Figure 2A) [47,48,49]. Autophagy-derived vesicles are necessary for cruzipain trafficking from the ER and the Golgi apparatus into the reservosomes. Fusion among these organelles augments the acidification of the luminal content of reservosomes and activates hydrolases to cleave stored lipids and proteins such as cruzipain by self-proteolysis. This process promotes reservosome maturation into lysosomes, allowing the amino acid and energy supply required to ensure parasite survival and differentiation [20,46]. Reservosomes are absent in MTs, whilst in epimastigotes, they occupy almost 6% of the total cellular volume [50,51].

Low levels of glucose seem to elicit a stress response by activating the signalling of metacaspases 3 and 5 (TcMA3 and TcMA5) (Figure 2A). TcMA3 arrests the cell cycle in the G1/S transition through a signalling pathway still uncovered, halting cellular division and promoting MTG [52]. Acting as a counterpart, also as a result of stress response, are heat shock proteins (HSPs). In *T. cruzi*, increased levels of the cochaperone stress-inducible protein 1 (STI1) were observed in stressed and adhered epimastigotes. TcSTI1 helps to form the complex between HSP70 and HSP90, and, although there is no difference in expression level between the epimastigote and the MT (Figure 2A), the levels of TcSTI1 increase in parasites during nutritional stress, followed by a substantial decrease in the MT form, showing an important regulation to maintain homeostasis and equilibrating the apoptosis response in *T. cruzi* [53,54].

In the context of nutritional deprivation, proteasomes have a crucial role. Parasites treated with lactacystin, a proteasomal inhibitor, presented impaired cellular growth and stopped in the G2 phase. Despite the halt of cellular replication, parasites present a substantial decrease in MTG [55]. Moreover, proteasomal activity varies during MTG, with the highest activity in the epimastigotes. In contrast, the ubiquitination profile remains the same, with variations in the level of oxidised proteins during MTG, indicating that ubiquitin-independent degradation by the proteasome acts as a possible regulator [56].

*Modulations in the nucleus*, such as epigenetic control and wide mechanisms of transcriptional and translational inhibition, are also deemed essential during MTG [57]. One piece of evidence is the decrease in the activity of RNA polymerase I and II as well as the compaction of kDNA observed throughout MTG [9,58], indicating a repression mechanism.

Chromatin remodelling is an important control to repress DNA replication and transcription in *T. cruzi* [59]. Histone deacetylases (HDACs), proteins capable of remodelling chromatin, catalyse the removal of acetyl groups, resulting in transcription inhibition and DNA replication by allowing the DNA to wrap itself more tightly on its histones. This reduces DNA accessibility to cellular machinery [60]. The use of HDAC inhibitors reduces MTG despite no difference in growth rate, implying that chromatin modulation is crucial for MTG [61]. Although the signalling and intrinsic mechanisms are yet to be found, the sirtuin TcSIR2RP1, a member of the HDACs, migrates from the cytosol to the nucleus, promoting deacetylation and, as a result, a reduction in transcription and replication (Figure 2B). TcSIR2RP1 overexpression increased MTG by 59% compared to the control group and enhanced infectivity (Figure 3) [62]. The existence of diverse points of posttranslational modifications (PTMs) was observed in the structure of the parasite’s histones, with global levels of histone acetylation and methylation varying during MTG. Acetylation levels increased by 55% in adhered cells, mainly at the 24 h and 48 h timepoints. Increasing levels of dimethylation of H3K76 were also observed during MTG, indicating that this PTM is relevant to the cell cycle [63].

Another component of chromatin control whose trigger is yet unknown is the *T. cruzi* High Mobility Group B (TcHMGB) protein (Figure 2B). TcHMGB acts as an architectural protein by inducing changes in the nuclear structure, and its overexpression, besides affecting many other aspects of the cell cycle, reduces MTG, corroborating that the chromatin structure is also an important point of regulation (Figure 3) [65].

DNA metabolism seems to also be implicated in MTG. The replication protein A (RPA) acts in DNA replication, repair, and recombination [76]. TcRPA is colocalised with replication sites in nuclei but not in the kinetoplast, indicating that it may have a role in the stabilisation of ssDNA during DNA replication (Figure 2B) [66]. TcRPA knockout results in the slowdown of replication and an increase in the rate of MTG. TcRPA is present mostly in the MT cytoplasm, in contrast with the nuclear spatialization in the epimastigote [66]. In accordance, knockout of the nuclear export signal halts TcRPA transport to the cytoplasm and impairs MTG [67]. Components of the translational machinery are also downregulated in MTs. The small-subunit (SSU) processome is a ribonucleoprotein required for the synthesis of the 18S rRNA [77], and in *T. cruzi,* the SSU processome protein Sof1p (TcSof1) is downregulated in MTs (Figure 2B) [78].

*Calcium metabolism.* Ca^2+^ is implicated in a plethora of functions and signal pathways, one example being the already-mentioned PDH, which is calcium-sensitive. The knockout of two of the three genes coding for inositol 1,4,5-trisphosphate receptors (IP3Rs), which form an ion channel that mediates Ca^2+^ release from the *T. cruzi* ER, leads to death. Both increases and decreases in the expression of TcIP3R affect MTG negatively, indicating that fine-tuning of the levels of calcium is required for normal cellular functioning [40,64]. TcCALPx11, a calpain-like protein from *T. cruzi*, has stage-specific regulation during MTG [68]. Parasite treatment with the calpain inhibitor MDL28170 results in the in vivo reduced capability of parasites to attach to *Rhodnius prolixus*’s midgut and a decrease in MTG (Figure 3) [69]. These studies indicate that calpains are stage-regulated and may be involved in parasite attachment, a crucial step for metacyclic differentiation, and, consequently, could be a key class of proteins for parasite infectivity.

*O-glycosylation* is an important posttranslational modification by which o-glycans are added to some proteins. Golgi UDP-GlcNAc:Polypeptide O-α-N-Acetyl-D-Glucosaminyl transferase 2 from *T. cruzi* (TcOGNT2) is involved in the biosynthesis of O-glycan, being downregulated in MTs compared to epimastigotes. TcOGNT2 overexpression decreases MTG without affecting proliferating epimastigotes. MTG inhibition by TcOGNT2 overexpression does not depend on its catalytic activity, suggesting that this protein may activate a signalling pathway still uncharacterised (Figure 3) [70].

*Posttranscriptional regulation* is regarded as one of the most important points of control during *T. cruzi* differentiation, considering this parasite does not seem to have transcriptional control elements such as promoter sites [79]. Currently, the best understood pathway is that of Tc-eIF2α (*T. cruzi* eukaryotic initiation factor-2α) (Figure 2B). Overexpression of an inactive mutated version of Tc-eIF2α, which is incapable of being phosphorylated, impairs MTG (Figure 3) [71]. In *T. cruzi*, blood-derived heme accumulated inside endosomes inhibits the action of TcK2 (*T. cruzi* eIF2α kinase) on the anterior portions of the midgut. When the parasite migrates to the hindgut, lower levels of heme prevent its cellular accumulation, resulting in the activation of TcK2. TcK2 phosphorylates Tc-eIF2α, inhibiting its capacity to exchange from the GDP-bound to the GTP-bound active form. This inhibition acts as a pro-apoptotic factor and reduces cellular levels of translation [72].

Amongst posttranscriptional modulation molecules, RNA-binding proteins (RBPs) are one of the best classes characterised in *T. cruzi*. One of the first RBPs indicated to be an MTG regulator was the zinc finger RBP TcZFP2, which is downregulated in MTs while the mRNAs to which it binds are upregulated (Figure 3) [73]. The knockout of TcZC3H31, another zinc-finger RBP expressed primarily in the forms present in triatomines, completely impairs MTG whilst not affecting epimastigote growth (Figure 3) [74]. On the other hand, TcZH3H12 knockout, another RBP, inhibits epimastigote proliferation and increases MTG by 20–30%, indicating it positively regulates genes involved in epimastigote growth whilst at the same time negatively regulating transcripts required for differentiation into MTs [80]. U-rich RBP protein TcUBP1 greatly increases after differentiation begins, enhancing MTG (Figure 3) [75]. A DEAD-box RNA helicase also acts at the mRNA level and is present in MTs at levels eight times higher than epimastigotes [81]. A shift in both temperature and the conditions of the medium causes alterations in the transcriptomic profile and in the mitochondrial levels of RNA, indicating that posttranscriptional regulation is triggered by sensing these changes in the environment [82,83]. Two RBPs present in the *T. cruzi* genome, RBP4 and DRBD8, are upregulated in MT [84].

## 5. Differently Expressed Proteins during MTG: Proteome Data

A fact highlighting posttranscriptional and posttranslational regulation is that the translatome resembles more closely the data collected by proteomics than transcriptomics during MTG [85]. Albeit 95% of transcripts are commonly shared by epimastigotes and MTs, only 67% of proteins are common to both stages. It has been estimated that 80% of the genes found to be lacking in the MT translatome are repressed. An analysis of translation efficiency on MTs—that is, genes with increased or decreased translation without alteration in the levels of mRNA—showed genes encoding members of the TS family were the most overrepresented, followed by proteases (especially GP63, GP82, GP85, and CRP), proteins related to the cytoskeleton, and RBPs [85]. In opposition, gene families that showed significant decreased translation efficiency include those encoding for ribosomal proteins, RNA polymerase I, genes related to protein synthesis such as those of hypusine, eIF5a, cyclin CYC2, and a homologue of the Silencing Function Protein (ASF1) [85]. Downregulation of the hypusine pathway may communicate with eIF5a downregulation since hypusine is necessary for eIF5a function, as verified in other eukaryotes [85].

This proteomic analysis correlates with previous studies on specific proteins important for MTG. A summary of the findings of differentially expressed proteins during MTG is shown in Figure 4. The proteomic analysis of MTG also found that protein expression patterns of intermediate forms of *T. cruzi* were more closely related to each other and to epimastigotes than to metacyclic trypomastigotes [86]. Two other important characteristics were found during MTG (phospho)proteomics: phosphorylation itself and the proteins differentially expressed in the membrane. During MTG, phosphorylation acts as a control for proteins involved in varied cellular processes, including nuclear proteins (transcription activator factor, NUP-1, and nucleosome assembly-related protein), surface protein DGF-1, cytosolic proteins (eEF-1α, eEF2, ATP-dependent RNA, RBPs, thiol-dependent reductase 1, and trypanothione synthetase), metabolism-related proteins (CAP5.5 and NAC), transporters (ABC), and dynein heavy chains [87]. Among surface protein groups, the ones found to be exclusively in the MT form include GP90, GP82, a subset of GP85, GP63 group I member b, ASP-2, TcMUCII, procyclic-form surface glycoprotein, and signal transduction proteins (a 24kDa flagellar calcium-binding protein, flagellar calcium-binding protein 3, rab7 GTP-binding protein, and a putative calcium-binding protein) [88]. In summary, the studies of proteomics during MTG seem to confirm that tight and intricate cellular signalling happens during this process.

## 6. MTG Results in Key Features of Host Cell Invasion

Molecules present on the MT surface are liable to adhesion and recognition by cell receptors, enabling either MT phagocytosis by immune cells or MT active invasion of non-phagocytic host cells. In vitro approaches have evidenced that MT invasion activates signal transduction pathways that culminate in Ca^2+^-dependent lysosome recruitment and exocytosis in the host membrane, contributing to PV formation [91,92,93,94], which is crucial for parasite internalisation, triggering amastigogenesis, and ensuring the parasite life cycle, as highly motile MTs cannot be retained in the intracellular milieu and thus replicate [94]. In this section, we will discuss the participation of several molecules involved in the intricate mechanisms leading to MT host cell invasion.

The glycoproteins GP82, GP90, GP30, and GP35/50 are the main parasite surface proteins known so far to be implicated in host invasion by MTs. They are attached to the parasite membrane by a glycosylphosphatidylinositol (GPI) anchor, susceptible to phosphatidylinositol-specific phospholipase C (PI-PLC) cleavage, and therefore can be released into the extracellular medium, where they can interact with host receptors [95]. Member of a multigene family belonging to the GP85/*trans*-sialidase superfamily, GP82 is a MT-specific molecule that successfully triggers cytosolic Ca^2+^ mobilisation, leading to actin disruption and lysosome recruitment (Figure 5) [96]. In oral infection murine models, GP82 is employed to selectively invade mucosal cell lines through binding to gastric mucin; it also directs MTs to bind to the underlying epithelial cells [97,98,99,100]. GP82 is expressed in differentiating intermediate forms of the parasite and reaches the membrane by colocalising with cruzipain [101], which will contribute to parasite internalisation through the degradation of fibronectin [102]. Experiments showed that the binding capacity of GP82 was inhibited by a nonselective beta-blocker propanolol, which also inhibited lysosome spreading [103]. The expression of gp82 is regulated by the interaction of UTR3′ with the RNA-binding protein UBP1, promoting longer expression in MT in comparison with epimastigotes [104,105]. GP82 competes for the cell receptors with other molecules expressed on the cell surface membrane of MTs that are released to downregulate invasion [105,106]. Even though GP82 is the major glycoprotein in the internalisation process, MTs deficient in GP82 can infect host cells through another type of glycoprotein, GP30, which can mediate cell entry, mobilise Ca^2+^, and, like GP82, be recognised by the monoclonal antibody 3F6, although it is not capable of mediating entry in mucosal cell lines [97,99].

During MT invasion, GP82 triggers parasite PLC activation that cleaves the membrane lipid phosphatidylinositol 4,5-bisphosphate (PIP2) into the second messengers diacylgycerol (DAG) and inositol 1,4,5-trisphosphate (IP3), leading to calcium release and protein kinase C (PKC) activation (Figure 5) [107,108]. Phosphatidylinositol 3-kinase (PI3K) and protein tyrosine kinase (PTK) activation result in phosphorylation of p175, a protein undetectable in non-infective epimastigotes [106], and under PTK inhibition, MT infectivity is reduced [109]. Concerning host cell signalling during invasion, the recognition of MT GP82 by LAMP2 induces the activation of phospholipase C, with the generation of products that contribute to PKC activation and the downstream ERK1/2 pathway (Figure 5) [110]. Also, GP82 induces mTOR dephosphorylation by the PI3K/PKC pathway and further translocation of TFEB, a transcription factor associated with mTOR and regulator of lysosome biogenesis, as demonstrated by MTs’ colocalisation with lysosome biomarkers LAMP2 and mTOR (Figure 5) [111]. mTOR’s involvement in MT internalisation was corroborated by the mTOR inhibitor rapamycin, which affects the phosphoinositide 3-kinase PI3K/PKC pathway and induces lysosomal concentration in the perinuclear region [111,112]. In addition, PKC downregulation inhibits lysosomal exocytosis and MT invasion [112,113].

Unlike GP82, GP90 downregulates parasite entry by impairing the binding of GP82 to the host cell [114] in a way where activity has an inverse association with infectivity [115]. GP90 is expressed and released at low levels by the CL strain (DTU-TcVI) and elevated levels by the G strain (DTU-TcI), the latter being known for having impaired invasion capacity. Even though GP90 binds more efficiently to HeLa cells than GP82, it triggers lower Ca^2+^ mobilisation and impairs lysosome spreading, maintaining lysosomes in the perinuclear area, as demonstrated by experiments with the recombinant GP90 protein [115,116]. It is present in differentiating forms of the parasite, which suggests that the presence of glycoproteins in the cell membrane is not stage-specific but rather a gradual increase in expression during MTG until they achieve the fully MT differentiated form [101].

Mucins are glycoproteins that present a dense array of *O*-linked oligosaccharides and are involved in cell adhesion, protection against proteolysis in the vector midgut, and parasite invasion [81,83]. They comprise a family of 500 to 700 genes characterised by repetitive regions of Thr_8_-Lys-Pro_2_ in tandem at the central region, a GPI-anchor at the C-terminal, and a signal peptide at the N-terminal. In *T. cruzi*, mucins are linked in the genome to trans-sialidase genes that might contribute to ensuring their coordinated expression and, thus, their intricate association with the sialylation process [117]. In this process, sialic acid, a crucial molecule for viability and propagation that is not synthesised by the parasite, is transferred from glycoconjugates in the mammalian cell to mucins on the parasite surface [117]. This sialic acid transfer, activated by a specific trans-sialidase during MT invasion, causes the assembly of the Ssp3 epitope, which is required for recognition by the host cell [118]. It has been shown that trans-sialidases activate invasion of the host cell using the “eat me” signal through G-coupled receptors at the epithelial cell synapse, binding glycans and promoting microparticle uptake [119].

The mucin-like gene family (TcMUG) comprises proteins with similar genetic characteristics as the TcMUC family but with shorter regions [117]. TcMUGs ranging from 35 to 50 kDa (GP 35/50) and enriched with sialic acid are the major representatives of MUGs on MTs. The removal of sialic acid from these proteins results in more infective MTs by increasing their ability to adhere to the host cell and trigger intracellular Ca2+ signalling [118]. It is suggested that GP35/50-mediated invasion requires F-actin recruitment via activation of adenylyl cyclase and cAMP production [120]. Mucin-associated surface proteins (MASPs) are also associated with entry into the host cell. MASP52 is secreted when MTs interact with the host’s cells and induce endocytosis [121].

Different isoforms of GP63 are present in all stages, being less abundant in MTs and epimastigotes compared to amastigotes. In MTs, GP63 lacks *N*-glycosylation and thus has a smaller size (55 kDa) than that of epimastigotes and tissue culture-derived trypomastigotes (61 kDa). In addition, GP63s from MTs are not found on the surface of the parasite, like in other forms, but are localised inside the parasite cell. Anti-GP63 antibodies were used to pre-treat MTs cultivated with myoblasts, and the results showed a reduction in infection, indicating GP63s’ involvement in host cell recognition through a yet unknown mechanism [122].

Mevalonate kinase (MVK) is an enzyme of the ergosterol biosynthesis pathway that is secreted by both amastigotes and MTs, interfering with host cell signalling pathways when bound to the host cell surface (Figure 5). In vitro, MVK treatment induces phosphorylation of the MAPK signalling pathway through activation of ERK 1/2 and p38. It also interferes with actin remodulation and has a bi-functional modulatory role in cell invasion. While in extracellular amastigotes, it increases internalisation, but in MTs, treatment with MVK negatively modulates invasion, inhibiting parasite internalisation [113].

Serine-, Alanine-, and Proline-rich proteins, also called SAPs, are a multigene family classified into four groups (SAP1 to SAP4) according to the presence of endoplasmic reticulum signal peptide and/or addition of glycosylphosphatidylinositol anchor [123]. SAPs are highly expressed in MTs and released into the extracellular medium by epimastigotes and MTs as soluble factors or as components of secreted vesicles. They bind to the host cell, triggering intracellular Ca^+2^ mobilisation, and, probably in synergy with GP82, induce exocytosis of the host cell lysosome during MT internalisation (Figure 5) [124].

## 7. Metacyclic Trypomastigotes Exhibit Key Defence Mechanisms to Avoid Host Immune Response

Although some *T. cruzi* MTs may be destroyed at the entry site by the host’s local innate immune factors, some of them can avoid host defence mechanisms before either being passively internalised by phagocytic cells or actively invading nucleated host cells. The initial site of parasite entrance can critically affect the host’s immune response [3]. In the case of vector transmission, *T. cruzi* MTs are favoured by the triatomine saliva released at the bite site, which contains several salivary anti-haemostatic and immunomodulatory components that may facilitate parasite transmission and help MT survival [125].

*T. cruzi* itself relies on an arsenal of surface proteins to attach to and invade host cells, leading to the formation of the PV. Once retained in the PV, MTs take advantage of the low pH to induce the activation of a set of proteins that support their survival and allow their escape from the PV into the cytoplasm. These proteins prevent their killing, trigger their differentiation into amastigotes, and enable their replication. Sequestering of sialic acids by *T. cruzi* trans-sialidases from lysosome-associated membrane proteins LAMP1 and LAMP2 promotes membrane weakening and PV disruption by Tc-tox, a protein analogous to host complement C9, released from the parasite’s surface [126,127,128,129].

Inside macrophages, dendritic cells, or neutrophils, all of which are professional phagocytes, MTs must be able to survive in a highly oxidative milieu. To kill the parasite, specific enzymes from the host, such as NADPH phagocyte oxidase (phox) and inducible nitric oxide synthase (iNOS), can produce reactive oxygen or nitrogen species (ROS or RNS), respectively [130]. All these reactive molecules and their intermediates are able to target a broad range of pathogen components retained in the PV, including thiols, metal centres, tyrosine proteins, DNA, and lipids [131], leading to death. Both ROS and RNS can mobilise iron, favouring reactive molecule generation and therefore intensifying its toxicity. ROS and RNS are effective components against MTs [132,133,134]. To be able to survive, MTs coevolved to produce antioxidant enzymes, such as superoxide dismutases (SODs), which neutralise the host’s oxidative stress response. MT SODs, such as TcSOD A, B1, B2, and C, are distributed in different compartments of the parasite, where they can neutralise superoxide [135]. Other enzymes, such as cytosolic tryparedoxin peroxidase (TcCPX) and mitochondrial tryparedoxin peroxidase (TcMPX), are able to neutralise the reactive molecule intermediates H_2_O_2_ and peroxynitrite [136,137]. *T. cruzi* ascorbate-dependent heme-peroxidase (TcAPX) can also inhibit H_2_O_2_ [138]. A proteomic analysis revealed that these enzymes are overexpressed in MTs compared to epimastigotes [86] and can be observed at increased levels in virulent strains when compared to attenuated ones, highlighting their role and importance in host infection. Iron mobilisation generated by ROS production due to MT invasion also favours amastigotes, making iron available for amastigote metabolism, which burdens amastigote replication [139].

The host complement system is one of the first lines of defence against protozoan parasites. *T. cruzi* MTs have also evolved to overcome its pathways to survive and establish infection. MTs induce the release of microvesicles from blood cells, which bind to the C3 convertase C4b2a on the *T. cruzi* surface, leading to the inhibition of both the classical and lectin complement pathways [140]. They are also able to bind to the host factor H, hence inhibiting C3b cleavage [141].

Another strategy to inhibit the classical and lectin complement pathways is performed by *T. cruzi* calreticulin (TcCRT), expressed on the surface of MTs, which can prevent host C4 interaction with MASP2 or C1s, thus inhibiting C4 conversion to C4b and blocking C3 convertase formation [142,143]. TcCRT also binds to C1q [117,127], which is a positive signal to trigger phagocytosis, assisting in the internalisation of MTs [142,144,145]. A protein named GP160, or complement regulatory protein (CRP), found on the surface of MTs, can bind to C3b and C4b [146,147]; another one, GP58/68, prevents factor B association with C3b linked to the membrane [148]; and *T. cruzi*’s complement C2 receptor inhibitor trispanning (TcCRIT), a 32 kDa protein homologous to the C4 beta-chain, can block C4–C2 interaction and the subsequent C2 cleavage by MASP2 or C1s [149,150]. Besides these proteins, MTs also present trypomastigote decay-accelerating factor (T-DAF), a surface glycoprotein of 87–93 kDa that acts like the host DAF, regulating C3 convertase formation by binding to C3b and competing with factor B [151,152,153,154]. Finally, transcriptomic data suggest that the initial contact of human dendritic cells with *T. cruzi* activates the virus response, leading to interferon-induced gene activation, and deciphering how the parasite modulates signalling cascades that may antagonise these pathways needs to be explored [155,156].

## 8. Concluding Remarks

Despite being well documented for more than a century, Chagas disease remains a challenge due to the coevolution of *T. cruzi* with its hosts, resulting in a parasite that has adapted the harsh conditions of nutrient depletion to its own gain. Although its life cycle is well understood, the connection between nutritional depletion, cAMP signalling, nuclear remodelling, posttranscriptional control, and programmed cell death is unknown. MTG is the bridge between the invertebrate and vertebrate hosts; understanding its mechanism makes possible the design of new drugs or even pesticides to fight Chagas disease. In addition, the advancement of bioinformatics and biotechnology tools can also help to resolve these loose ends. The question that remains to be solved is: is there a sudden trigger that causes cell remodulation or the activation of multiple signalling pathways is necessary? If so, is there a certain necessary order for this activation? Despite all the evidence, MTG, a fundamental process that enables the parasite to successfully infect the host, remains an unsolved puzzle.

## Figures and Tables

**Figure 1 ijms-25-00117-f001:**
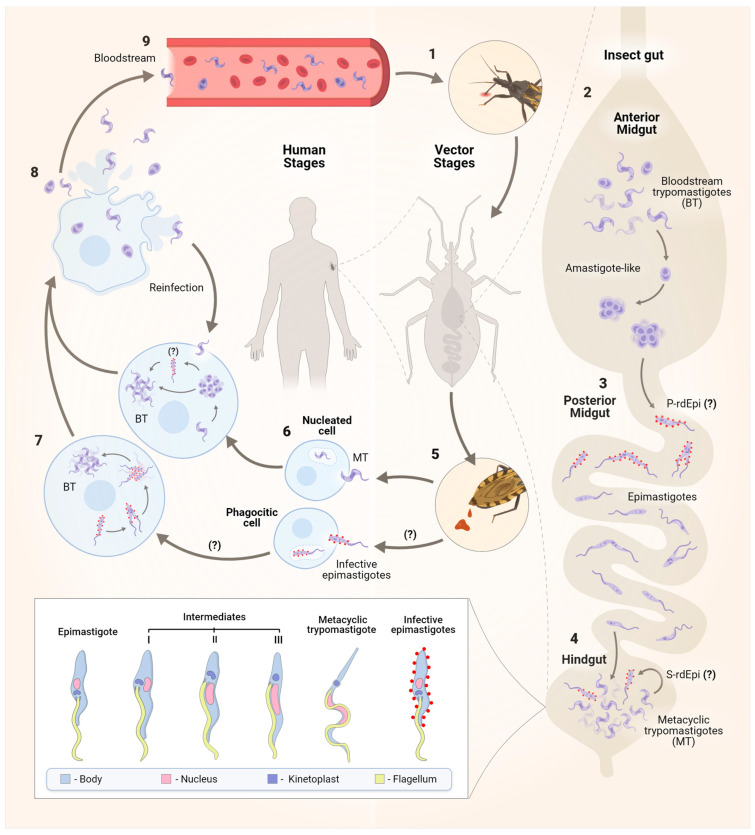
The life cycle of the protozoan *Trypanosoma cruzi*. The parasite has a dixenous life cycle, infecting both insects from the Reduviidae family and mammals. After blood feeding (1), in the insect anterior midgut (AM), surviving pleomorphic forms of *T. cruzi* differentiate into amastigote-like forms in the first hours of AM infection (2). These forms then migrate to the posterior midgut (PM), where primary epimastigogenesis (P-rdEpi) occurs, giving rise first to replicating infective epimastigotes and subsequently to epimastigotes that colonize the PM (3). The latter migrate to the hindgut, where MTG occurs, and depending on the nutritional milieu status, MTs also differentiate into infective epimastigotes in a process called secondary epimastigogenesis (S-rdEpi) (4). MT forms [15], infective epimastigotes [16], and epimastigotes are excreted in the faeces (5). Resistant forms to the host complement and other innate immune system components can infect various nucleated host cells, including macrophages, muscle cells, epithelial cells, fibroblasts, and nerve cells. Both MTs and infective epimastigotes enter host cells through distinct mechanisms, ultimately leading to the formation of the parasitophorous vacuole (PV) (6). After escaping from the PV, MTs differentiate into amastigotes, which replicate in the cytosol of the infected cell (7) and may transition through the rdEpi form before completing their transformation into bloodstream trypomastigote (BT). Trypomastigotes and, to a lesser extent, amastigotes rupture the host cell (8), enter the bloodstream, and then gain access to other cells to initiate a new cycle of division (9). The signal (?) refers to steps of the cell cycle that still need to be better established.

**Figure 2 ijms-25-00117-f002:**
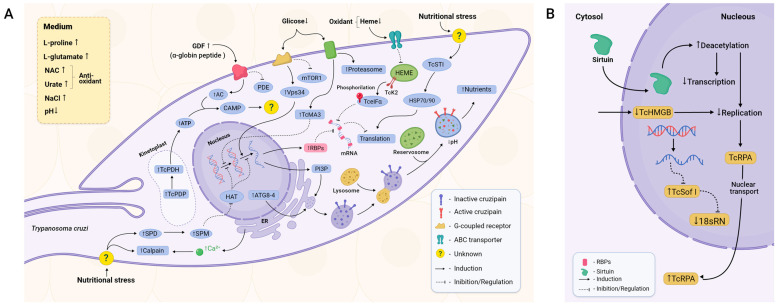
Main components of metacyclogenesis triggering. (**A**) MTG depends on a poor environment, stress-related signalling, and autophagy. Peptides derived from digested blood and low levels of glucose and heme trigger intracellular signalling pathways mediated by cAMP, calpains, metacaspases, and other molecules such as spermidine in a generalised stress-related cellular signalling. As a consequence, apoptosis-like responses, autophagy, and cell arrest are elicited. Posttranscriptional and posttranslational factors reduce universal levels of protein expression, favouring specific sets of proteins that confer tolerance to stress, such as pH reduction and temperature shift. They also induce maturation of reservosomes, enabling the parasite to survive in this unfavourable environment. Virulence factors that help the parasite to evade the vertebrate host immune response and augment its ability to invade cells are synthesised, completing its transition from the non-infective replicative epimastigote form to its infective non-replicating MT form. (**B**) Once sirtuins such as TcSIR2RP1 are transported into the nucleus, deacetylation levels are increased, decreasing both transcription and DNA replication. On the other hand, proteins related to DNA replication, such as TcRPA, migrate from the nucleus to the cytoplasm, halting DNA replication. The RNAs required for ribosome assembly are also downregulated, leading to impairment of the translational machinery itself.

**Figure 3 ijms-25-00117-f003:**
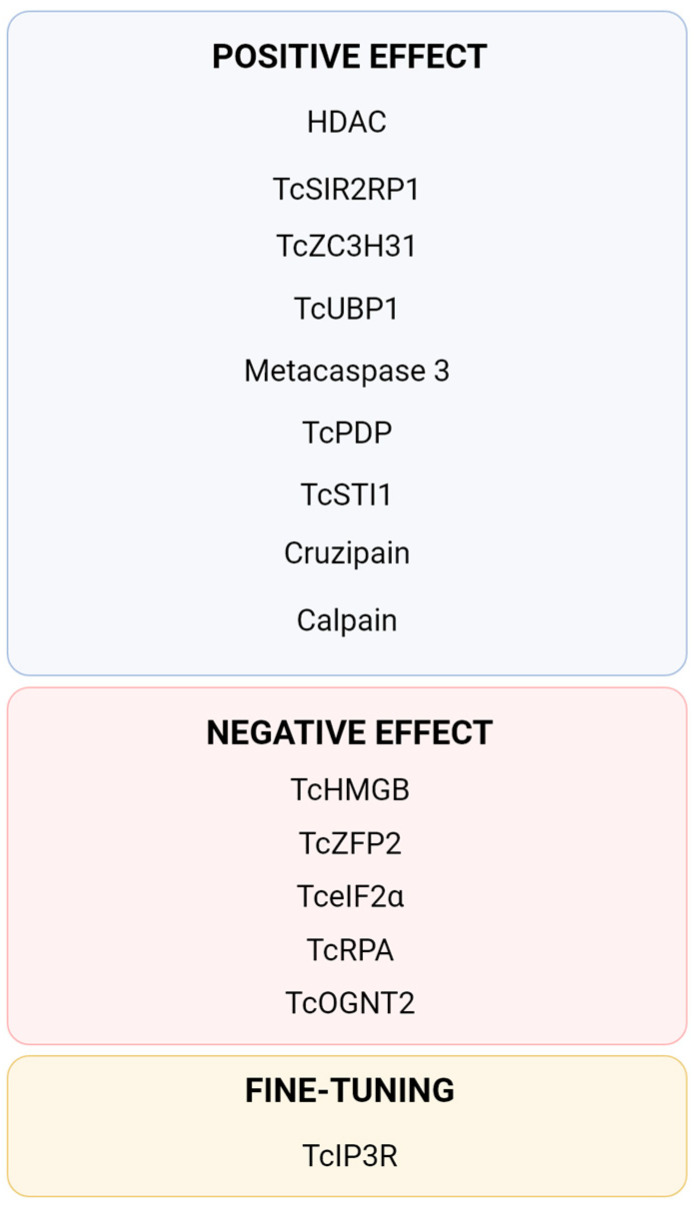
Summary of currently described proteins directly involved in the process of *Trypanosoma cruzi* metacyclogenesis. TcPDP [40], TcIP3R [40,64], Cruzipain [47,48,49], Metacaspase 3 [52], TcSTI1 [53,54], HDAC [60,61], TcSIR2RP1 [62], TcHMGB [65], TcRPA [66,67], Calpain [68,69], TcOGNT2 [70], Tc-eIF2α [71,72], TcZFP2 [73], TcZC3H31 [74] and TcUBP1 [75].

**Figure 4 ijms-25-00117-f004:**
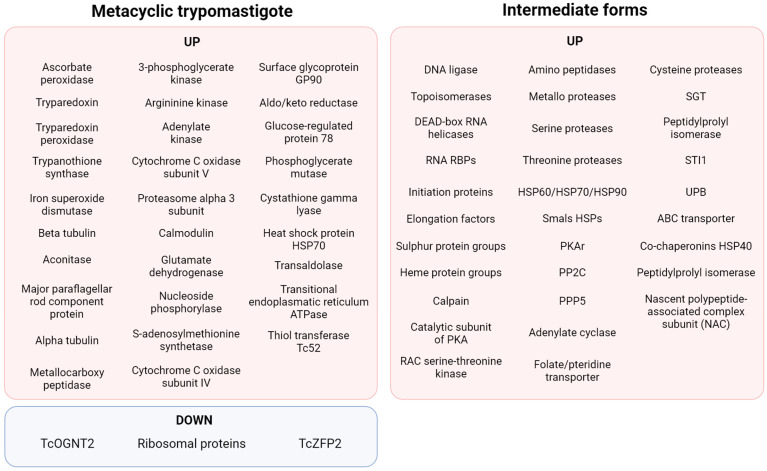
Summary of the proteomic findings regarding MTG [87,89,90].

**Figure 5 ijms-25-00117-f005:**
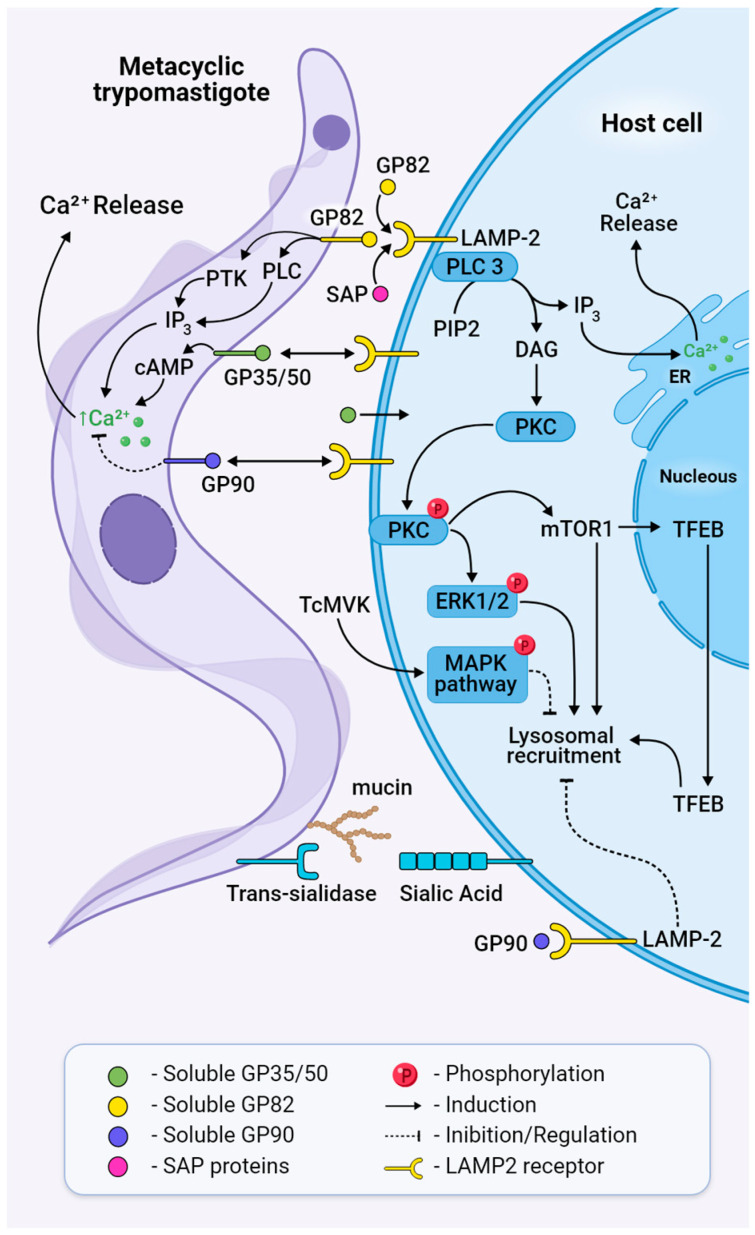
Bidirectional signalling pathways activated in both MT and the host cell during invasion. Trans-sialidases transfer sialic acid from the host cell membrane to mucins or mucin-like glycoproteins on the surface of the parasite, a process necessary for parasite adhesion and penetration. GP82 binds to its receptor LAMP2 and triggers cleavage of PIP2 through the activation of PLC, generating IP3 and DAG, leading to Ca^2+^ release to the cytosol and PKC translocation to the membrane. Phosphorylated PKC activates ERK1/2 and mTOR dephosphorylation, leading to TFEB translocation, actin disruption, and lysosome recruitment. Treatment with MVK negatively modulates invasion by inhibiting parasite internalisation through the phosphorylation and activation of MAPK. Extracellular SAPs bind to LAMP2 and activate cytosolic Ca^2+^ mobilisation. Ultimately, this will result in the formation of the parasitophorous vacuole and host cell invasion. GP90 acts as a downregulator of invasion and binds to the host cell without triggering a Ca^2+^ transient.

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
