# Peer review of "Metacyclogenesis as the Starting Point of Chagas Disease"

_ijms, 2023, doi:10.3390/ijms25010117_

Round 1

Reviewer 1 Report

Comments and Suggestions for Authors

The manuscript by Ferreira et al is well-written, structured, and concise.  However, I cannot see any significant contribution to the field. All the information presented has been summarized, presented, and discussed in many reviews in the field. I cannot see how the publication of this manuscript could provide a better understanding or summary of the MTG in T. cruzi.

Author Response

Thank you very much for taking your time to review our manuscript. Although there are several reviews exploring diverse aspects of Chagas disease and the biology of the parasite, as far as we know, there has been no review in the last five years focusing solely on the metacyclic form. The vast majority of studies provide information on experiments carried out with culture-derived trypomastigotes. This is most likely due to the difficulty of working with the metacyclic form in vivo in the hematophagous vector, as well as the difficulty of differentiating and cultivating this form in vitro. Our review provides a thorough survey of the complex process of metacyclogenesis known so far, which involves cyclic adenosine 3′,5′-monophosphate (cAMP) signaling, autophagy/apoptosis, pathways, calcium metabolism, nuclear remodeling and post-transcriptional regulation. Another distinguishing feature of this review is the gathering of proteins differentially expressed by metacyclic forms revealed by proteomic studies, that provide insights to the readers into potential virulence factors, drug targets and/or vaccinal antigens. In addition, we present an update on process of metacyclic trypomastigote interaction with the host cell and the signaling pathways activated in both cells, as well as the mechanisms for evading the host's defenses. Finally, we consider that our review was exhaustive and focused on the life form that is key to the transmission of the parasite and the establishment of Chagas disease, which is a field of research that deserves to be further explored.

Reviewer 2 Report

Comments and Suggestions for Authors

I have a few editorial comments:

Line 116 (end) /117 (beginning). Authors should reconcile these two lines.

Line 179. Authors should consider if they need to expand the term ‘DTU’, as currently this is the only mention in the text.

Line 459. Authors should add the DTUs (lineages) of CL Brener and G strains, as this may be an important consideration related to the differing levels of Gp90 expression.

Fig 3. Authors should add citations for the proteins listed.

END

Author Response

Line 116 (end) /117 (beginning). Authors should reconcile these two lines.

  • Thank you for this observation. We deleted the line 117 because is a truncated sentence that begin in the next paragraph.

Line 179. Authors should consider if they need to expand the term ‘DTU’, as currently this is the only mention in the text.

  • We described the term in the text.

Line 459. Authors should add the DTUs (lineages) of CL Brener and G strains, as this may be an important consideration related to the differing levels of Gp90 expression.

  • We added the corresponding DTUs in the text.

Fig 3. Authors should add citations for the proteins listed.

  • We added the citations in the legend of figure 3

Reviewer 3 Report

Comments and Suggestions for Authors

The MS “Metacyclogenesis as the starting point of Chagas disease” provides an excellent review being able to summarize a big amount of literature and provides a figure of excellent quality.

Some improvements are required:

Abstract: the Abstract includes too much generalist information on the Chagas disease, authors need to rewrite completely including your main results. As is now written is not informative for the reader. Start with the information on line 18: This review provides……………..cells” and after you provide or your results and conclusions.

A material and methods section is required, in this section need to details with criteria you used to select literature included in this review (interval of years, languages of literature considered, …).

 Page 3 line 116 “In the insect vector, it was observed that both” is a "floating sentence" in the MS, should be a mistake during a process of “cut and paste” when you were preparing your MS

 REFERENCES section is plenty of format mistakes, for example reference 3 Trypanosoma cruzi should be in italics. Reference 15 should not be in capital letters. I recommend to take time to recheck the format of literature section.

Author Response

The MS “Metacyclogenesis as the starting point of Chagas disease” provides an excellent review being able to summarize a big amount of literature and provides a figure of excellent quality.

-We really appreciate for taking your time to review our manuscript.

Some improvements are required:

Abstract: the Abstract includes too much generalist information on the Chagas disease, authors need to rewrite completely including your main results. As is now written is not informative for the reader. Start with the information on line 18: This review provides……………..cells” and after you provide or your results and conclusions.

A material and methods section is required, in this section need to details with criteria you used to select literature included in this review (interval of years, languages of literature considered, …).

  • As this is a literature review and not a systematic review, we have summarized in the abstract general information on the topic along with the main subjects to be covered. And since it's not a systematic review, we consider it would not be applicable/appropriate to divide the abstract into methodology, results and discussion. Likewise, it would be difficult to establish criteria such as dates, as we have presented information from Carlos Chagas' first publication in 1909, as it is still extremely relevant and accurate, to the present day.

 Page 3 line 116 “In the insect vector, it was observed that both” is a "floating sentence" in the MS, should be a mistake during a process of “cut and paste” when you were preparing your MS

  • Thank you for this observation. We deleted the line 117 because is a truncated sentence that begin in the next paragraph.

 REFERENCES section is plenty of format mistakes, for example reference 3 Trypanosoma cruzi should be in italics. Reference 15 should not be in capital letters. I recommend to take time to recheck the format of literature section.

  • References were checked and reformatted.

Round 2

Reviewer 1 Report

Comments and Suggestions for Authors

None